# Environmental and Sensor Integration Influences on Temperature Measurements by Rotary-Wing Unmanned Aircraft Systems

**DOI:** 10.3390/s19061470

**Published:** 2019-03-26

**Authors:** Brian R. Greene, Antonio R. Segales, Tyler M. Bell, Elizabeth A. Pillar-Little, Phillip B. Chilson

**Affiliations:** 1School of Meteorology, University of Oklahoma, 120 David L. Boren Blvd, Ste 5900, Norman, OK 73072, USA; tyler.bell@ou.edu (T.M.B.); epillarlittle@ou.edu (E.A.P.-L.); chilson@ou.edu (P.B.C.); 2Center for Autonomous Sensing and Sampling, University of Oklahoma, 120 David L. Boren Blvd., Ste 4600, Norman, OK 73072, USA; tony.segales@ou.edu; 3Advanced Radar Research Center, University of Oklahoma, 3190 Monitor Ave. Norman, OK 73019, USA; 4School of Electrical and Computer Engineering, University of Oklahoma, 110 W. Boyd St., Ste 150, Norman, OK 73019, USA

**Keywords:** UAS, sensor integration, thermistor, sensor placement, observations, sensor calibration

## Abstract

Obtaining thermodynamic measurements using rotary-wing unmanned aircraft systems (rwUAS) requires several considerations for mitigating biases from the aircraft and its environment. In this study, we focus on how the method of temperature sensor integration can impact the quality of its measurements. To minimize non-environmental heat sources and prevent any contamination coming from the rwUAS body, two configurations with different sensor placements are proposed for comparison. The first configuration consists of a custom quadcopter with temperature and humidity sensors placed below the propellers for aspiration. The second configuration incorporates the same quadcopter design with sensors instead shielded inside of an L-duct and aspirated by a ducted fan. Additionally, an autopilot algorithm was developed for these platforms to face them into the wind during flight for kinematic wind estimations. This study will utilize in situ rwUAS observations validated against tower-mounted reference instruments to examine how measurements are influenced both by the different configurations as well as the ambient environment. Results indicate that both methods of integration are valid but the below-propeller configuration is more susceptible to errors from solar radiation and heat from the body of the rwUAS.

## 1. Introduction

In the past 10–15 years, atmospheric scientists and engineers around the world have begun to address the shortage of planetary boundary layer (PBL) observations [1,2] by leveraging the rapid miniaturization of thermodynamic sensors and accessibility of open-source autopilot technologies to develop integrated unmanned aircraft systems (UAS, also referred to as remotely piloted aircraft systems, RPAS) for atmospheric research (e.g., [3,4,5,6,7,8,9,10,11,12,13]). Decades of advances in manned aircraft research have largely benefited the development of fixed-wing UAS (fwUAS) in particular (e.g., [14,15,16]), which has led to a focus on the use of fwUAS in the early years of atmospheric research with UAS [3,4,5,7,8,17]. Another advantage of fwUAS is their flight times, which are typically around an hour. This makes utilizing fwUAS advantageous when needing to cover large horizontal transects of the environment or in performing several vertical profiles without needing to land to refuel or recharge. In addition to fwUAS, rotary-wing UAS (rwUAS) are becoming more commonly utilized because of their improved commercial accessibility, versatility, and quasi-Euclidean perspective [11,18,19,20,21,22].

One common rwUAS application in atmospheric sciences is through collecting thermodynamic and kinematic measurements akin to traditional radiosondes: temperature, pressure, relative humidity, and horizontal wind vector as a function of altitude [13,19,20,23]. For thermodynamic measurements, it is essential for a sensor to be shielded from solar radiation and aspirated with the ambient environment to avoid influences from non-environmental heating and decoupling with the environment [24,25,26]. Solar radiation can bias temperature measurements in excess of 1 ${}^{\circ }$C in weak ambient winds [26]. Additionally, thermistors can be susceptible to self-heating when current is continually run through them and can build up heat. This bias can be mitigated with proper ventilation across the sensor [27,28,29].

Tower-mounted thermistors typically mitigate this issue by shielding their sensors in chambers aspirated either mechanically with a fan or by the ambient wind [27,28,29]. Radiosondes rely on passive ventilation from the relative airflow during flight [30,31]. However, implementing these methods is nontrivial when integrating sensors with an rwUAS. While utilizing a fan can ensure adequate aspiration, it may also negatively affect flight endurance by increasing take-off weight and power consumption. Studies have demonstrated that it is possible to aspirate sensors using the air currents produced by propellers on an rwUAS [32]. However, several non-environmental heating sources, such as the rwUAS motors and frictional heating on the propeller tips, can affect measurements in this region without careful considerations for sensor placement [32,33]. Recent numerical simulations of rotor downwash have mainly focused on analyzing flow dynamics and the aerodynamic implications on aircraft design (e.g., [34]), yet little work has been done to understand the impact of rotor wash on thermodynamic measurements. Studies have also shown that sensor placement and aspiration on a UAS can impact the response time of their measurements [35].

Obtaining characteristic temperature measurements of the convective boundary layer (CBL) in space and time is often a challenge [2,36], as turbulent eddies range in size from millimeters to kilometers and act to transport heat, moisture, and momentum [37,38]. Many commercially available temperature sensors utilized in conjunction with UAS are capable of sampling at 1–10 Hz, which can resolve some of these turbulent processes. These measurements therefore can exhibit large variations temporally, even when taken at a quasi-fixed spatial position on a hovering rwUAS. Validation against tower-based measurements can be difficult due to spatial variability in surface fluxes, especially across differences in land-surface characteristics (e.g., [39]). Another way to assess the quality of UAS temperature measurements is by quantifying the variability between measurements from different redundant sensors onboard the aircraft. This approach allows for the systematic evaluation of whether any differences that exist are characteristic of the environment or are biases from the platform itself.

When utilizing an rwUAS to collect thermodynamic atmospheric observations, an optimal sensor-UAS integration must be achieved to obtain characteristic measurements. This study will outline this optimization process using observations from varying sensor layouts on an rwUAS that seeks to answer the following questions:To what extent do the different configurations bias measurements?How does incoming solar radiation and the angle relative to the oncoming wind affect observations?Do the benefits to data quality (if any) using a fan offset the reduced flight time from additional weight and power consumption?

This paper will be structured in the following manner: in Section 2, the rwUAS and its different sensor configurations used in this experiment will be introduced and the experimental methods will be described. In Section 3, results will be presented in the context of the three questions posed above. In Section 4, these questions will be directly addressed with any associated caveats. Finally, a summary and conclusions from this experiment will be presented in Section 5.

## 2. Materials and Methods

### 2.1. CopterSonde

The experiments in this study utilized the CopterSonde rwUAS (Figure 1), a quadcopter developed and built by the Center for Autonomous Sensing and Sampling (CASS) at the University of Oklahoma (OU) for atmospheric research. This CopterSonde is the second design iteration of the original rwUAS described by Greene et al. [32]. The vehicle frame is based on a modified version of the Lynxmotion HQuad500 (Swanton, VT, USA) and has four fixed-pitch rotors mounted at the end of four arms that are attached to the main body. The CopterSonde utilizes four carbon fiber 11 in (27.9 cm) diameter and 5.5 in/rev (14.0 cm/rev) pitch T-style propellers, each driven by a T-motor U7 700 kV electric motor and a Lumenier 30 A BLHeli32 4-in-1 speed controller (Sarasota, FL, USA). The flight endurance is about 18 min with a payload of 0.4 kg and a total all-up weight of 2.3 kg. The CopterSonde has a top flight speed of 26.4 m s${}^{-1}$ and can be flown safely in wind speeds of up to 21 m s${}^{-1}$.

The CopterSonde is equipped with a Pixhawk Cube 2.1 autopilot board (Hex Technology, Sha Tin, Hong Kong), which was used as the main controller for flight stabilization, navigation and operation. This autopilot system is capable of real-time kinematics (RTK) differential GPS (DGPS), which allows for positional accuracy on the order of centimeters. The Pixhawk runs open-source ArduCopter algorithms, which have been customized to add compatibility with the desired atmospheric sensors and allows for collected data to stream to the ground station over telemetry in real time. Additionally, a custom wind estimation algorithm was developed and implemented by measuring the inclination of the drone in a horizontally steady position. The algorithm then sends commands to the autopilot to turn the drone in the yaw axis until the roll is minimized and the drone is tilted only in pitch axis. By maintaining a constant cross-section of the rwUAS pointed into the oncoming wind, methods of calculating horizontal wind speeds based on the aircraft’s attitude [23] have demonstrated greater accuracy than without enabling the wind estimator.

For ease of calibration and comparisons, this version of the OU CopterSonde was designed to have a modular payload with customizable configurations of thermodynamic and chemical sensors. This payload is mounted on the detachable main body shell, which is 3D printed with a composite plastic material. These sensors are capable of running independently of the main rwUAS body, and vice versa. For collecting thermodynamic measurements, the CopterSonde is outfitted with a suite of three iMet-XF PT 100 thermistors (International Met Systems, Grand Rapids, MI, USA) and three HYT 271 capacitive relative humidity sensors (Innovative Sensor Technology, Ebnat-Kappel, Switzerland). These sensors interface with the autopilot system for onboard synchronization with sampling rates of 10 Hz.

### 2.2. Sensor Placement

For a direct comparison of different methods of aspirating the sensors, the CopterSonde in this study was modified to have two sets of the thermodynamic sensor suites discussed in Section 2.1, for a total of six temperature and six relative humidity sensors. One set was mounted inside cylindrical plastic solar shields underneath the propellers, and the other set was integrated inside a ducted fan as part of the interchangeable payload on the front end of the aircraft (Figure 2; Table 1).

The location of the sensors underneath the propellers (herein referred to as the “arm propeller” or “AP” sensors) were selected based on the considerations from Greene et al. [32], which ideally avoids heat from the motors and from frictional heating at the tips of the propellers. This setup was also utilized by the previous version of the CopterSonde and is an effective solution for aspirating thermodynamic sensors without having to design and build a custom rwUAS.

Commonly, tower-mounted sensors ensure data quality through the use of aspirated chambers. However, reduced flight times from the added weight and power consumption from implementing this type of system on an rwUAS has left this configuration relatively unexplored, and the majority of systems utilize the under-propeller aspiration method. Because of the potential benefits to data quality, the CopterSonde was designed with sensors mounted inside an L-duct aspirated by a fan and attached to the front of the aircraft (“front fan” or “FF” configuration). This fan is calibrated to draw air across the sensors at a constant speed of 12 m s${}^{-1}$ with considerations for how the geometry of the bend affects the air speed. This setup also allows for the sensors to all be collocated in a relatively more homogeneous environment as compared to sensors under multiple different propellers.

To assess and compare the performances of these two methods of sensor aspiration, a series of experiments were conducted, which will be outlined in Section 2.3.

### 2.3. Field Experiments

Experiments were conducted at the Kessler Atmospheric and Ecological Field Station (KAEFS) in Purcell, OK, USA, which consists of a mixed grass prairie ecosystem that is characteristic of the United States Southern Great Plains region. It is the site of the Washington Oklahoma Mesonet 10 m tower (WASH; Figure 3d), which is one of 120 sites across the state of Oklahoma that collects regular 1-min observations of temperature, relative humidity, pressure, solar radiation, wind speed, and wind direction [27,28]. To emphasize the effect of oncoming wind relative to the rwUAS orientation on thermodynamic measurements, a suitably windy day was chosen to collect observations. On 13 September 2018, the weather at KAEFS was typical of a late warm season afternoon with temperatures in the upper 20s Celsius, southerly winds at 3–6 m s${}^{-1}$, and partly cloudy with intermittent periods of sunshine (Figure 4). In a series of seven independent trials, the CopterSonde was programmed to hover at 10 m above ground level (AGL) directly adjacent to the tower for 10–15 min while measuring temperature and relative humidity in the scoop and shields simultaneously. The 9 m temperature sensor (Figure 3d) is aspirated by the ambient winds, so the accuracy of measurements increases with wind speed. At this height and ambient wind conditions, the tower measured air temperature with an accuracy of ±1 ${}^{\circ }$C, wind speed accurate within ±0.3 m s${}^{-1}$, and wind direction accurate within ±3${}^{\circ }$. In each trial, the only parameters altered were the utilization of the CopterSonde’s automatic wind estimation algorithm and its heading relative to the oncoming wind. Of the seven trials, the CopterSonde faced into the wind for three of them (Figure 3a), and perpendicular to the wind in the remaining four (Figure 3b,c; Table 2). This setup allowed for observations to be considered with regards to sensor location, angle relative to wind, and incoming solar radiation, which will be discussed in the following sections.

### 2.4. Analysis Techniques

To address the three questions posed in this study, analysis will first consist of bulk statistics comparing the performance of the sensors with respect to each other (Section 3.1). For consistency and to allow for direct comparisons, all measurements from the FF and AP configurations were averaged to a common 1-s grid in time. This interval was selected based on the approximate response time of the thermistors in their respective setups. One method to evaluate the overall effects of sensor location on temperature measurements is by comparing the amount of variation between sensors for the two configurations. One method of determining the spread in temperature sensor measurements is by calculating the mean of absolute differences (MD), which is given by:(1)$$MD\left(t\right)=\frac{|T_{1}\left(t\right)-T_{2}\left(t\right)|+|T_{1}\left(t\right)-T_{3}\left(t\right)|+|T_{2}\left(t\right)-T_{3}\left(t\right)|}{3},$$
where MD is the time series of mean of absolute differences of temperature measurements, and T_1_–T_3_ are simultaneous measurements from sensors 1–3 at time *t*. MD was calculated for every time step from all flights separately for both sensor configurations to compile two overall distributions for comparison. This metric was chosen because it summarizes the relative precision between the individual temperature sensors. Since there is no true reference measurement in these experiments, precision was chosen as the metric of interest since it will provide insight to a UAS operator, whose configuration (FF or AP) can give the most consistent results.

For a baseline reference, MD was also calculated for the six sensors from when they underwent calibrations in the Oklahoma Mesonet instruments laboratory. The arm propeller sensors were calibrated freestanding in a temperature-controlled chamber ranging in temperatures from −40–60 ${}^{\circ }$C, changing by 10 ${}^{\circ }$C every ten minutes. In this commercial off-the-shelf chamber, the sampling volume was mechanically stirred for aspiration across the sensors. The FF sensors along with the scoop were calibrated in a separate, larger temperature-controlled chamber ranging in temperatures from 10–30 ${}^{\circ }$C, changing by 10 ${}^{\circ }$C every hour. The ducted fan was on during these calibrations to continually aspirate the sensors. Individual linear temperature sensor biases were obtained by comparing 1-min averaged measurements to the reference chamber National Institute of Standards and Technology (NIST) traceable sensor temperatures and computing their average differences. The distributions of MD from these calibrations (taking into account the sensor biases) therefore offer an ideal sense of the spread in measurements (i.e., precision) from the sensors in question.

To draw conclusions on the MD distributions based on statistical confidence intervals, the bootstrapping method was determined to be the most appropriate, as this approach makes no assumptions regarding the underlying population or distribution shape [40]. For this analysis, 10,000 resamples of the distribution were performed to determine the 95% confidence interval of the resampled medians. If the 95% confidence intervals from different distributions do not overlap, the difference in their medians are statistically significant.

After the bulk statistics are discussed, analysis will then focus on results from time series of CopterSonde and Mesonet tower observations to identify relationships between rwUAS measurements and varying environmental conditions (Section 3.2 and Section 3.3).

## 3. Results

### 3.1. Differences between Shield and Scoop

A direct comparison of the MD distributions for both the AP and FF configurations in the field (all points from flights 1–7) and laboratory (calibration chambers; Figure 5) reveals several notable features. For these distributions, field measurements are averaged over 1-s intervals to be consistent with their response times. Although the calibrations were performed based upon 1-min averages, the 1-s intervals are utilized in this statistical analysis for more appropriate comparison.

The entire distribution of MD during the FF sensor calibrations lies below 0.09 ${}^{\circ }$C, with an interquartile range (IQR) of 0.06 ${}^{\circ }$C. This indicates that these sensors are highly precise under steady-state conditions with constant ducted fan aspiration. Precision in this sense and for the rest of this study will be relative to the other sensors, i.e., low variability among a collection of simultaneous measurements. Additionally, 95% of MDs during the calibration of the AP thermistors were below 0.12 ${}^{\circ }$C with an IQR of 0.05 ${}^{\circ }$C. This calibration distribution is likely skewed higher than the FF calibration MDs since the AP sensors underwent a larger range of temperatures, but the same argument generally holds: the AP thermistors are also highly precise while aspirated under steady-state conditions.

By contrast, for all of the observations collected outside in a CBL, the variations in measurements between the sensors increases. Regardless of incoming solar radiation and relative wind speed, the median MD of measurements from the FF configuration is only 0.08 ${}^{\circ }$C. While the center of this distribution is approximately the same as that in the lab, the 5th–95th percentile range of MDs in the field is more than double the lab range. In practice, this amount of spread is relatively small and is a reasonable amount of precision decrease when compared to the lab values. Furthermore, the median MD between the AP sensors during the flights is 0.10 ${}^{\circ }$C, a small increase over that from the FF setup. However, the AP sensor distribution exhibits a 5th–95th percentile range of 0.22 ${}^{\circ }$C, a considerable increase over the FF distribution. These variations in temperature measurements border on being a systematic issue for overall accuracy and precision of the UAS measurements at the upper end of this distribution. In general, though, this configuration still measures temperature with an acceptable spread between sensors.

Because the 95% bootstrap confidence intervals of the FF and AP distributions in the field do not overlap (Figure 5), it is reasonable to conclude that these distributions are significantly different. This implies that, at any time, a set of measurements from the sensors in the AP configuration will have larger variability between them than the set of measurements from the sensors in the FF setup.

### 3.2. Effects from Solar Radiation

The first environmental influence related to sensor integration that will be addressed is solar radiation. Flights 1, 4, and 5 experienced sky conditions ranging from mostly sunny to mostly cloudy that correlated with responses in CopterSonde temperature measurements, and will be the focus of analysis.

During flight 1, the CopterSonde was oriented directly into the wind (Figure 3a) for a flight analysis time of about five minutes. The sky cover transitioned from mostly sunny during the first two minutes to mostly cloudy for the remaining period (Figure 6). For this initial sunny period, temperature measurements in the AP configuration had a spread of up to 0.60 ${}^{\circ }$C, whereas the FF sensors had a maximum spread closer to just 0.30 ${}^{\circ }$C. Sensors T2_RAP_ and T3_RAP_ tended to observe warmer temperatures in this period than sensor T1_LAP_ as well as all three FF sensors. At this time of day, the sun had an azimuth angle of 241${}^{\circ }$and an elevation angle of 40${}^{\circ }$. Because the CopterSonde maintained a heading of about 200${}^{\circ }$ in this timeframe, the sun would have been shining directly on the right side of the aircraft. With wind speeds between 4–6 m s${}^{-1}$, it is possible that the CopterSonde was tilted into the wind enough such that the solar shields housing the AP sensors were not adequate protection from the sun. Additionally, the shield itself would have absorbed some of the solar radiation as heat. A combination of these effects likely explain why the right two sensors recorded higher temperatures than the left.

Further supporting the hypothesis of bias from solar heating, it was observed that the incoming solar radiation decreased by more than half as the sun became obscured by cumulus clouds about three minutes into flight 1. Almost immediately following, the spread in temperatures from the AP sensors diminished to less than 0.30 ${}^{\circ }$C, and the FF sensor spread dropped slightly to about 0.20 ${}^{\circ }$C. With less direct sunlight in this period, all six sensors converged to match nearly identically with the Mesonet 9 m temperature observations, and the environment cooled by about 0.3 ${}^{\circ }$C. Overall results from this flight indicate that, with wind speed and direction relative to the CopterSonde, incoming solar radiation may potentially bias temperature sensors exposed to the sunlight if not properly shielded.

During the entire duration of flight 4 (Figure 3b), solar radiation remained below 200 W m${}^{-2}$ due to mostly cloudy sky cover (Figure 7). During the 7-min flight window, temperatures at 9 m fell by 0.8 ${}^{\circ }$C according to the Mesonet tower. After the CopterSonde spent around 30 s adjusting orientation to be perpendicular to the wind for this trial, all six of the thermistors converged to a spread of less than 0.1 ${}^{\circ }$C.

Throughout flight 5, the CopterSonde was oriented 146${}^{\circ }$ (into the oncoming wind), and the sun had an azimuth angle of 254${}^{\circ }$ at an elevation of 26${}^{\circ }$ (Figure 3a). The heading of the CopterSonde relative to the sun therefore mostly shaded sensor T1_LAP_ in its shield, whereas the shield for sensors T2–T3_RAP_ were exposed to direct sunlight with a clear sky. This differential heating across the AP sensor shields likely is the driver behind why T2–T3_RAP_ recorded warmer temperatures than T1_LAP_ and T1–T3_FF_ for the duration of the flight (Figure 8), as the illuminated shield did not effectively block the radiation from heating up its interior. Furthermore, sensors T1–T3_FF_ and T1_LAP_ detected a temperature drop of 0.3 ${}^{\circ }$C in less than 30 s just prior to 10:33 p.m. UTC also detected in the 1-min Mesonet temperature measurements. However, sensors T2–T3_RAP_ responded slower to this environmental change than the other CopterSonde sensors, likely due to the bias from solar heating.

Because of the differential solar forcing observed in flight 5 along with the weak solar forcing during flight 3 and the combination of forcings from flight 1, it has been demonstrated that sensors in the AP configuration are susceptible to appreciable bias due to solar radiation. Otherwise, they can perform just as well as sensors in the FF setup if not exposed to direct sunlight.

### 3.3. Effects from Wind Direction

In addition to solar radiation, it is possible that wind direction relative to the orientation of an rwUAS in flight may affect temperature observations. To test this potential, flights 6 and 7 will be analyzed.

During both flights 6 and 7 (Figure 3c), the sky was clear and the sun was setting with an elevation angle of 22${}^{\circ }$ decreasing to 18${}^{\circ }$ and an azimuth angle ranging from 258–261${}^{\circ }$. While the surface normal solar radiative flux was weak (less than 300 W m${}^{-2}$) relative to its daily maximum, the direct sunlight was still enough to drive intermittent turbulence as indicated by the wide variations in temperature measurements and diminishing wind speeds from flight 6 to 7 (Figure 9 and Figure 10). For these two flights, the CopterSonde maintained average yaw angles of 264${}^{\circ }$ and 241${}^{\circ }$, respectively, which was both perpendicular to the wind and almost directly at the sun.

Because of the similarities in the environment and experimental setup, the overall trends in CopterSonde temperature observations from flight 6 were generally reproduced in flight 7. Sensor T1_LAP_ on the left arm of the CopterSonde consistently displayed a cool bias of 0.10–0.20 ${}^{\circ }$C in comparison to sensors T2–T3_RAP_ on the right arm. The FF sensors maintained a smaller variance throughout, but occasionally sensor 3 also diverged with a cold bias of 0.10–0.20 ${}^{\circ }$C.

These results support the notion that the difference between the AP sensors on the left and right arms are a result of the orientation of the rwUAS with respect to the ambient wind. Since the left sensor was upwind from the body of the CopterSonde, it was measuring airflow after only interacting with the front left propeller, whereas sensors T2–T3_RAP_ were downwind of the main body. Previous studies (e.g., [32]) have shown that an rwUAS can modify the thermodynamics of the environment in which it is sampling by spreading heat from motors and frictional heating from spinning propellers. Additionally, because of the direct sunlight, it is possible that the grey plastic shell of the main body was absorbing heat that was then transported towards sensors T2–T3_RAP_ by the relative ambient wind. Based on the observations from flights 6 and 7, it is likely that non-environmental heating from these sources propagating downwind were detected by the AP sensors while the upwind sensors were relatively less susceptible.

The trend of FF sensor 3 occasionally exhibiting a cool bias during flights 6 and 7 was likely related to effects of solar radiation discussed previously in Section 3.2. The CopterSonde was oriented directly at the setting sun, which due to the geometry of the L-duct partially illuminated the inside. A laboratory test with a flashlight confirmed that direct sunlight was capable of reaching sensors T1–T2_FF_, but sensor T3_FF_ was low enough inside the duct that it was unlikely to be illuminated by the sun. Therefore, this is another example of the potential for solar radiation to influence temperature measurements if a sensor is not properly shielded.

## 4. Discussion

Based on the results presented previously in Section 3, it is apparent that the FF configuration on the CopterSonde is more successful than the AP setup at mitigating effects from solar radiation and rwUAS orientation relative to the ambient wind. As was shown in Section 3.3, however, the FF setup at the time was not immune to the effects of solar radiation. Due to the complex geometry of how an rwUAS in-flight is oriented in three-dimensional space, it is a nontrivial engineering challenge to design an ideal solar shield. Sensors fully immersed in concentric plastic ducts have been shown to be an effective way to dynamically aspirate the sensors while shielding them from the sun [41], but this configuration can increase the response time of the system by more than order of magnitude. This trade-off unfortunately makes an FF design similar to this an unrealistic solution for the needs in vertical resolution of observations required from rwUASs. Furthermore, it is possible that the aircraft and solar shield composition (e.g., carbon fiber and 3D-printed plastics) can have an impact on the measurements observed. Further studies will need to be done to examine potential benefits of integrating materials such as metamaterials and polarizers to reduce effects from solar radiation.

Aspiration of sensors from propellers still can be a viable solution in most cases, especially with weak solar heating. Utilizing the wind estimation algorithm in the rwUAS autopilot is capable of mitigating the effects of orientation with respect to the ambient wind. On average, however, this can still introduce biases from nonuniform solar heating across the aircraft as shown in Section 3.2. It is possible to employ more elaborate solar shielding than what was used in this study, but the trade-off between effectiveness and weight added can eventually become an issue here. This of course will vary on an individual basis, but the possibility still remains.

Another factor to consider when siting sensors on the arms of a multirotor UAS is the potential for increased vibrations by the torquing of the arms that can affect temperature measurements. These effects are likely muted in the FF configuration, which is subject only to the main body forces of the rwUAS. The extent to which this affects observations, however, is yet to be quantified.

Although not explicitly shown in this study, it is likely that measurements from the AP configuration may have larger biases in a stably stratified environment as compared to the FF. This is due to the propellers drawing air downward from an an unknown distance that is then measured by the sensors, whereas a ducted fan can draw in air from the horizontal. Further studies will need to be conducted in stable regimes to explore this concept of rwUAS-modified-environments further.

It is also possible that the FF sensors may have interacted with some of these heating sources, but further analysis with computational fluid dynamics (CFD) would be required to improve confidence in this scenario. Flights 2 and 3 (Appendix A) demonstrated levels of variability that may be a result of microscale fluctuations around the aircraft, which would also be a point of investigation with CFD simulations. Preliminary computer-aided design (CAD) simulations have alluded to temperature distributions inside of the L-duct near the fan, but more elaborate experiments will need to be conducted to make realistic conclusions.

## 5. Conclusions

In this study, the environmental effects on the quality of measurements from different thermistor configurations in an rwUAS were examined. An OU CopterSonde rwUAS was flown at a hover next to the Washington Oklahoma Mesonet tower at 10 m a total of seven separate times on 13 September 2018. This rwUAS was outfitted with six total PT 100 thermistors, three of which were aspirated in a ducted fan on the front of the aircraft, and the other three located in solar shields underneath the front two propellers. For each flight, the orientation of the CopterSonde relative to the environmental wind was varied. From the observations collected, the questions posed in Section 1 can be sufficiently addressed:In general, the atmospheric measurements obtained from thermistors in the FF configuration exhibit higher precision and accuracy than those in the AP. These differences are statistically significant, although the overall variances from both configurations are still relatively small from a data quality standpoint.Solar radiation can bias sensors appreciably when not properly shielded, with MDs greater than 0.20 ${}^{\circ }$C. The orientation of the rwUAS with respect to the ambient wind can also bias measurements when non-environmental heat sources (e.g., motor heat, frictional heating on the propeller, and heat on the main body when exposed to direct sunlight) propagate towards the downstream temperature sensors.Overall, the FF configuration is less susceptible to environmental influences than the AP configuration, which has been demonstrated both statistically and on a case-by-case basis. The impact on flight time from the added weight and power consumption of a ducted fan has proven to be less significant than expected with the current operational version of the CopterSonde, which has only been limited in maximum altitude by visual line-of-sight restrictions. When the resources allow, the authors recommend implementing the ducted fan setup for collecting thermodynamic observations with an rwUAS. Otherwise, it is imperative to be cognizant of the potential biases introduced when aspirating sensors with propeller wash.

Results from this study have laid the framework for new designs of the OU CopterSonde, which will likely incorporate a ducted fan that is more isolated from solar radiation. Furthermore, simulations of propeller downwash in CAD will need to be conducted to determine if heat from an rwUAS main body can influence temperature measurements when the aircraft is not oriented directly into the wind. Finally, quantification of the extent to which an rwUAS modifies the vertical structure of the environment is an important consideration in sampling the PBL. Future research is required in these areas to better understand the significance of thermodynamic measurements collected with rwUAS.

## Figures and Tables

**Figure 1 sensors-19-01470-f001:**
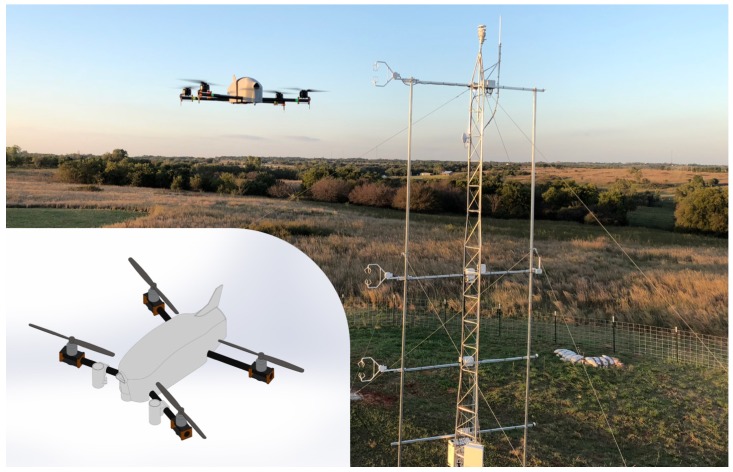
Photograph of the OU CopterSonde rwUAS (Norman, OK, USA) in flight next to an experimental 10 m flux tower, and a 3D model (inset) of the aircraft in flight. The 3D model design is the one used in this study, which includes sensors located in solar shields underneath the front two propellers and housed in a ducted fan on the front of the main body. The solar shields and main body with the ducted fan were 3D printed in white and grey plastic, respectively.

**Figure 2 sensors-19-01470-f002:**
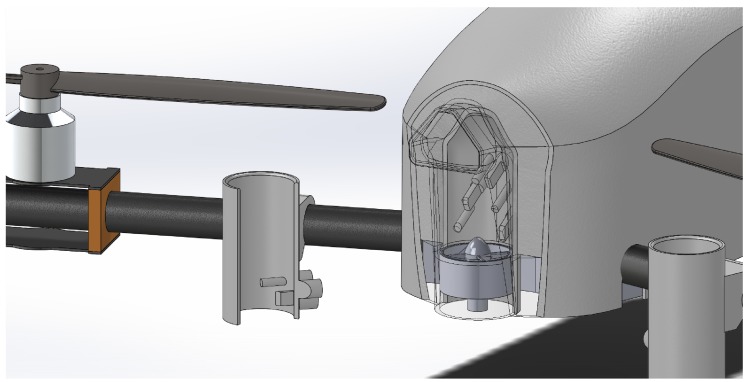
3D model cross-section of the OU CopterSonde rwUAS highlighting the dual configuration sensor placements. This perspective highlights the location of the individual sensors inside the cylindrical plastic solar shields under the propellers (AP) and the ducted fan on the front of the main body (FF).

**Figure 3 sensors-19-01470-f003:**
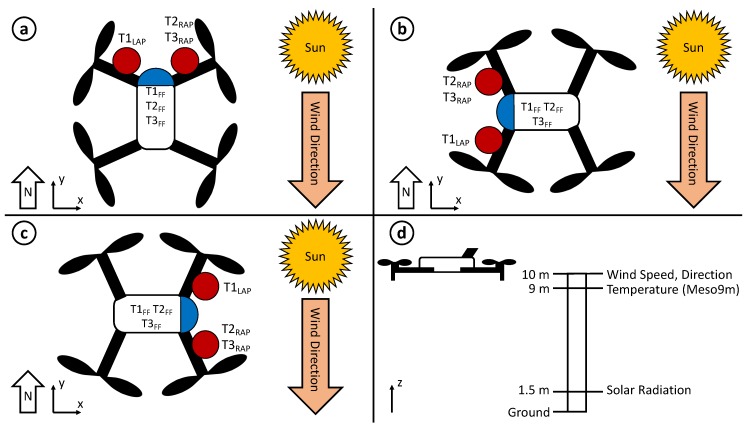
Schematic detailing the orientations of the CopterSonde relative to the sun and oncoming wind for experiments: (**a**) 1, 3, and 5; (**b**) 2 and 4; and (**c**) 6 and 7; (**d**) the CopterSonde hovered at 10 m for each experiment directly adjacent to the Washington Mesonet tower, which had sensors located at 1.5, 9, and 10 m (figure dimensions not to scale).

**Figure 4 sensors-19-01470-f004:**
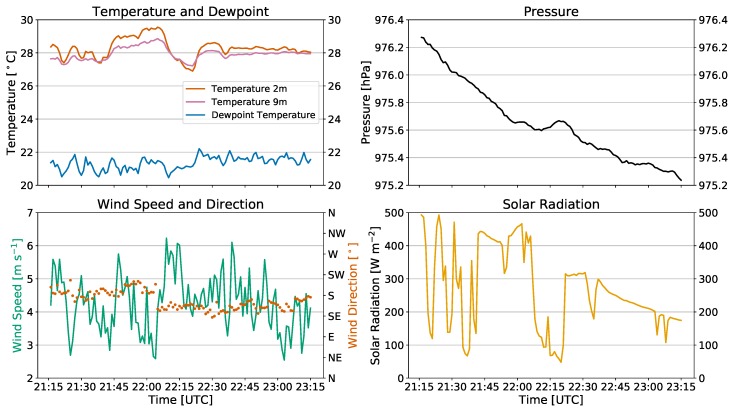
Time series of 1-min temperature and dewpoint temperature (**top left**), pressure (**top right**), wind speed and direction (**bottom left**), and solar radiation (**bottom right**) observations for the afternoon of 13 September 2018 from the Mesonet site in Washington, OK, USA. The time range depicted encompasses the beginning of the first flight through the end of the last flight.

**Figure 5 sensors-19-01470-f005:**
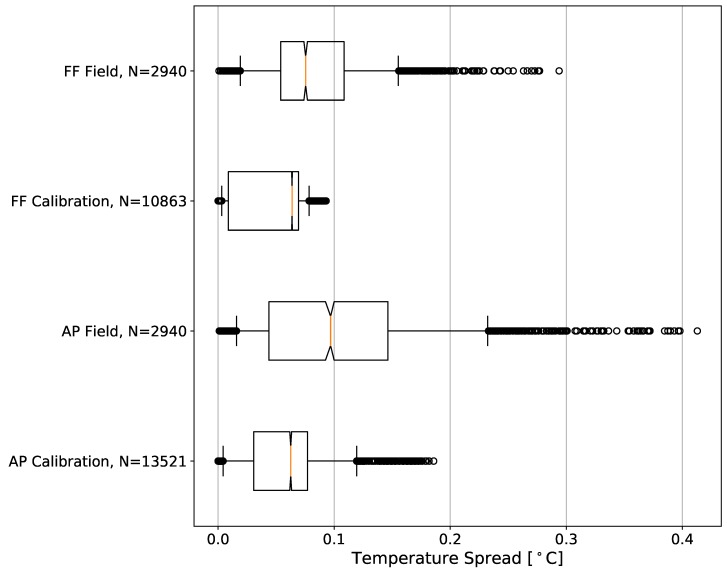
Notched box and whisker plots of temperature mean of absolute differences [${}^{\circ }$C] distributions from the different sensor configurations calculated using Equation (Equation 1). The 25th, 50th, 75th are represented by the boxes, and the 5th and 95th percentiles are represented by the whiskers. Samples outside the 5th–95th percentiles are plotted individually as circles. The notches estimate the 95% confidence interval of the median using a bootstrap method with 10,000 resamples.

**Figure 6 sensors-19-01470-f006:**
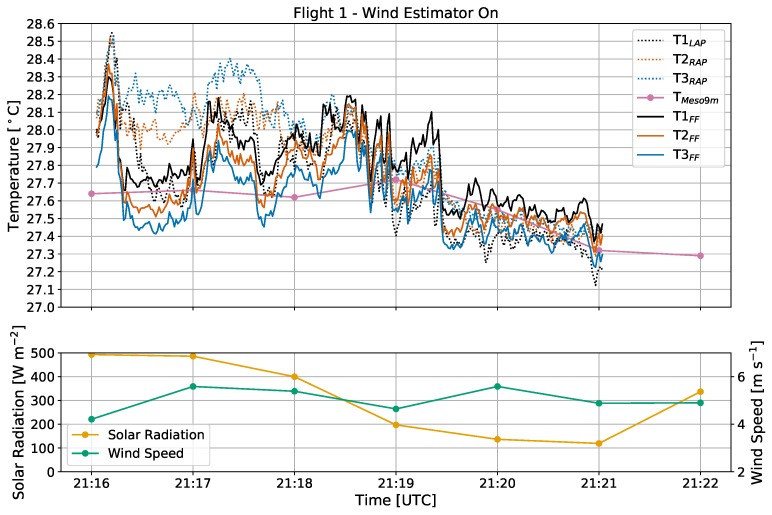
Time series of temperature [${}^{\circ }$C], solar radiation [W m${}^{-2}$], and wind speed [m s${}^{-1}$] from flight 1. The wind estimator was on, and the CopterSonde was oriented directly into the oncoming wind. In the top figure, temperatures from the front fan setup are in solid lines, whereas measurements from the arm propellers are dotted. Measurements from the Mesonet tower collected at 1-min intervals and plotted with connected solid circles.

**Figure 7 sensors-19-01470-f007:**
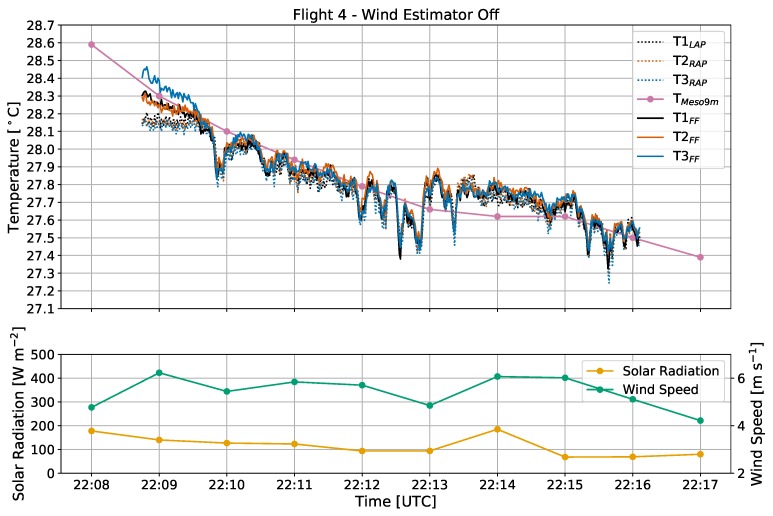
Time series of temperature [${}^{\circ }$C], solar radiation [W m${}^{-2}$], and wind speed [m s${}^{-1}$] from flight 4, with the same labeling conventions used as in Figure 6. The wind estimator was off and the CopterSonde was oriented 90 degrees counterclockwise from the oncoming wind and away from the sun.

**Figure 8 sensors-19-01470-f008:**
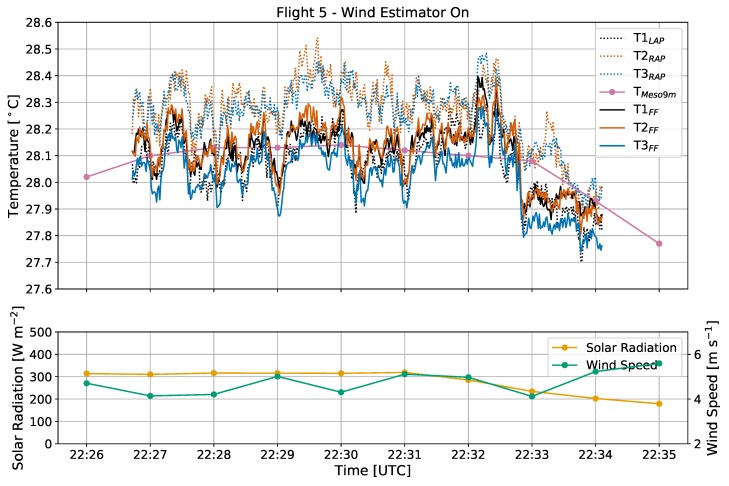
Time series of temperature [${}^{\circ }$C], solar radiation [W m${}^{-2}$], and wind speed [m s${}^{-1}$] from flight 5, with the same labeling conventions used as in Figure 6. The wind estimator was on and the CopterSonde was oriented directly into the oncoming wind.

**Figure 9 sensors-19-01470-f009:**
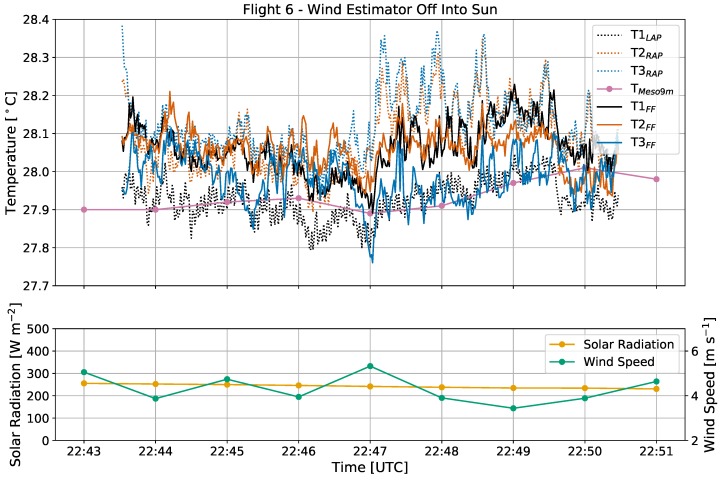
Time series of temperature [${}^{\circ }$C], solar radiation [W m${}^{-2}$], and wind speed [m s${}^{-1}$] from flight 6, with the same labeling conventions used previously. For this flight, the wind estimator was off and the CopterSonde was oriented 90 degrees *clockwise* from the oncoming wind, which was towards the sun.

**Figure 10 sensors-19-01470-f010:**
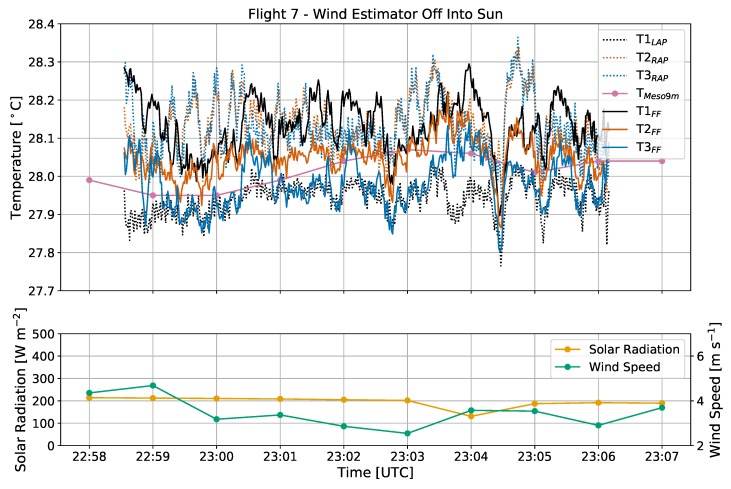
Time series of temperature [${}^{\circ }$C], solar radiation [W m${}^{-2}$], and wind speed [m s${}^{-1}$] from flight 7, with the same labeling conventions used previously. As with flight 6, the CopterSonde flew perpendicular to the wind facing the sun.

**Table 1 sensors-19-01470-t001:** Summary of temperature measurement sources.

Name	Sensor Model	Location and Aspiration	Solar Protection
Front Fan	iMet-XF	Ducted Fan	Plastic L-duct
(FF)	PT 100	Ducted Fan	
Left Arm Propeller	iMet-XF	Left Front Arm	Cylindrical Plastic Shield
(LAP)	PT 100	Propeller Wash	
Right Arm Propeller	iMet-XF	Right Front Arm	Cylindrical Plastic Shield
(RAP)	PT 100	Propeller Wash	
Mesonet	RM Young 41342	Mesonet Tower at 9 m	10-plate Radiation Shield
(Meso9m)	Platinum RTD	Ambient Wind	

**Table 2 sensors-19-01470-t002:** Description of the 7 CopterSonde flights on 13 September 2018 at KAEFS. The wind was primarily from the south and southeast for the duration of these flights (Figure 4). When the wind estimator was “off”, the CopterSonde yaw angle was controlled manually.

Flight #	Wind Estimator	Heading	Orientation	Sky Cover
1	On	S	Into wind	Sunny then mostly cloudy
2	Off	E	Away from sun	Mostly cloudy then sunny
3	On	S	Into wind	Mostly sunny
4	Off	E	Away from sun	Mostly cloudy
5	On	S	Into wind	Clear
6	Off	W	Into sun	Clear
7	Off	W	Into sun	Clear

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
