# Peer review of "Environmental and Sensor Integration Influences on Temperature Measurements by Rotary-Wing Unmanned Aircraft Systems"

_sensors, 2019, doi:10.3390/s19061470_

Reviewer 1 Report

The content of this paper is novel and significant. 

High quality of presentation and easy to read.

however, lack of enough mathmatical validation for the experiments and algorithm.  In Section 2.4, there need more theoritical support for the paper. 

Author Response

The authors would like to thank Reviewer 1 for their inputs to this manuscript. The reviewer comments will be repeated in black, italics, and bold, with the author responses in red.

The content of this paper is novel and significant. 

High quality of presentation and easy to read.

Thank you for these remarks.

however, lack of enough mathmatical validation for the experiments and algorithm.  In Section 2.4, there need more theoritical support for the paper. 

Further explanation of the MD metric has been provided in section 2.4, with more context and justification for why it was chosen. The equation for MD was elaborated upon to demonstrate how it is a time series of values that is calculated for each time step. This should therefore strengthen the results and conclusions presented.

Reviewer 2 Report

Your paper was nicely presented, and you fully cover all the requirements for good scientific journal. All your processes, methods, models and results were clearly defined. Your results are solid, your calibration process is properly documented. The writing was clear, and it was very easy to follow all your ideas. I think this is a good introduction of your method and idea. But there are several issues that I am interested to see how you can fix in the next iteration of your work. Your main task will be reducing the effects of the solar radiation effects in the sensors. I think you did not consider the material of your UAS chassis enough. Just using any plastic won’t do. You need to do further study into the material that can properly isolate your sensor from your system biases and external noise. There are very interesting ways of radiative cooling with metamaterials. Metamaterials can be 3D printed and added to the frame of your UAS. There are also polarization materials that can be embedded into your drone to reduce the effects of the solar light. Also, vibrations affect the electronics in your system. You might want to consider looking into this phenomenon. Is probable that the sensors under the propellers had more mechanical vibrations than the ones in the front. That can be another reason to added noise into your results between the AP and FF. Adding a damping mechanism can help too. Your work is a good initial proof of concept that can evolve into a more elegant solution to a very interesting problem. You documented good lessons learn from the initial design that can be improved and better validate your method. I am looking forward for the next stage of your research.

Author Response

The authors would like to thank Reviewer 2 for their thoughtful inputs to this manuscript. The reviewer comments will be repeated in black, italics, and bold, with the author responses in red.

Your paper was nicely presented, and you fully cover all the requirements for good scientific journal. All your processes, methods, models and results were clearly defined. Your results are solid, your calibration process is properly documented. The writing was clear, and it was very easy to follow all your ideas. I think this is a good introduction of your method and idea.

Thank you for these remarks.

But there are several issues that I am interested to see how you can fix in the next iteration of your work. Your main task will be reducing the effects of the solar radiation effects in the sensors. I think you did not consider the material of your UAS chassis enough. Just using any plastic won’t do. You need to do further study into the material that can properly isolate your sensor from your system biases and external noise. There are very interesting ways of radiative cooling with metamaterials. Metamaterials can be 3D printed and added to the frame of your UAS. There are also polarization materials that can be embedded into your drone to reduce the effects of the solar light. Also, vibrations affect the electronics in your system. You might want to consider looking into this phenomenon. Is probable that the sensors under the propellers had more mechanical vibrations than the ones in the front. That can be another reason to added noise into your results between the AP and FF. Adding a damping mechanism can help too. Your work is a good initial proof of concept that can evolve into a more elegant solution to a very interesting problem. You documented good lessons learn from the initial design that can be improved and better validate your method. I am looking forward for the next stage of your research.

These are very interesting ideas that we are certainly interested in evaluating further. Additional comments in the discussion section have been added to suggest the possibility for (1) vibrations on the arms of the rotorcraft and (2) exploring the ideas of metamaterials and polarizers to reduce the effects of solar radiation.

Reviewer 3 Report

This is a relevant, timely and very well-conceived paper, at least for the UAS community that I much enjoyed reading.

The outline and structure of the paper follows a clear logic and guides the reader readily through the paper. The introduction provides a solid background and motivation and clearly states the objectives and key questions addressed by the study. The section on “Materials and Methods” provides sufficient information on the methodology employed and the experimental details of the investigations. Results are thoroughly discussed and critically evaluated. Finally, the conclusions provide convincing answers to the three key questions posed and give some suggestions for future work.

The text is well written and easy to comprehend. The illustrations and tables are well designed and serve to clarify the content of the paper in an appealing manner.

While the results and conclusions are derived for a specific rwUAS and specific instrumentations, they do provide most useful information for similar studies and experimental setups.

Overall, I do not find any significant issues that would need improvements and therefore unreservedly recommend publication of the paper in Sensors.

Author Response

The authors would like to thank Reviewer 3 for their thoughtful insights on this manuscript. The reviewer comments will be repeated in black, italics, and bold, with the author responses in red.

This is a relevant, timely and very well-conceived paper, at least for the UAS community that I much enjoyed reading.

The outline and structure of the paper follows a clear logic and guides the reader readily through the paper. The introduction provides a solid background and motivation and clearly states the objectives and key questions addressed by the study. The section on “Materials and Methods” provides sufficient information on the methodology employed and the experimental details of the investigations. Results are thoroughly discussed and critically evaluated. Finally, the conclusions provide convincing answers to the three key questions posed and give some suggestions for future work.

The text is well written and easy to comprehend. The illustrations and tables are well designed and serve to clarify the content of the paper in an appealing manner.

While the results and conclusions are derived for a specific rwUAS and specific instrumentations, they do provide most useful information for similar studies and experimental setups.

Overall, I do not find any significant issues that would need improvements and therefore unreservedly recommend publication of the paper in Sensors.

Thank you for all of these comments.